**Subject Category:**
Biology (whole organism)

behaviour/ecology

maternal care, post-weaning care, social environment, kin selection, predation

**Author for correspondence:**
Angela R. Freeman
e-mail: arf86@cornell.edu

# Gone girl: Richardson's ground squirrel offspring and neighbours are resilient to female removal

Angela R. Freeman[1,2], Thomas J. Wood[2], Kevin R. Bairos-Novak[2,3], W. Gary Anderson[2] and James F. Hare[2]

[1]Department of Psychology, Cornell University, Ithaca, NY, USA
[2]Department of Biological Sciences, University of Manitoba, Winnipeg, Manitoba, Canada
[3]Department of Biology, University of Saskatchewan, Saskatoon, Saskatchewan, Canada

ARF, 0000-0001-5103-5674; KRB-N, 0000-0002-0152-1452;
JFH, 0000-0001-6408-8484

Within matrilineal societies, the presence of mothers and female kin can greatly enhance survival and reproductive success owing to kin-biased alarm calling, cooperation in territory defence, protection from infanticidal conspecifics, joint care of young and enhanced access to resources. The removal of mothers by predators or disease is expected to increase the stress experienced by offspring via activation of their hypothalamic-pituitary-adrenal axis, increasing circulating glucocorticoids and reducing offspring survival and reproductive success. Yet, few studies have removed mothers in the post-weaning period to examine the assumed physiological and fitness consequences associated with these mortality events. We examined how the loss of a mother affects juvenile Richardson's ground squirrels' (*Urocitellus richardsonii*) faecal glucocorticoid metabolites and their survival. Given that neighbours are often close kin, we further hypothesized that conspecific removal would similarly diminish the fitness of neighbouring individuals. Upon removing the mother, we detected no impact on offspring or neighbouring conspecific faecal glucocorticoid metabolites in the removal year, or on overwinter survival in the following year. Furthermore, no impact on neighbour reproductive success was detected. Given the high predation rates of ground squirrels in wild populations, resilience to a changing social environment would prove adaptive for both surviving kin and non-kin.

# Significance statement

The importance of maternal care in social rodents is well understood in that young are reliant on mothers for milk and warmth; however, we know little about the impacts of mothers on offspring in the immediate post-weaning period when offspring are often considered self-sufficient. We studied the effect of mother removal on her offspring and show that her removal had no detectable effect on the concentration of stress-related faecal glucocorticoid metabolites immediately after removal, or on offspring overwinter survival in the subsequent year. We also examined effects on adult neighbours and detected no stress response, survival or reproductive impacts on related and unrelated neighbours. Thus, our study reveals that Richardson's ground squirrels are particularly resilient to a changing social environment, perhaps due to the high levels of predation experienced by members of this species.

# 1. Introduction

Living in close social groups has substantial fitness impacts. Association with conspecifics can improve fitness through increased survival rates [1], improved predator detection and defence [2], and improved reproductive success from sharing resources, territory or offspring care [3,4]. Living in groups can also impose costs, however, including increased competition for mates and resources, inbreeding, reproductive suppression, parasite and disease transmission, and detection by predators [3,5–8]. Sociality, therefore, is thought to evolve in species where the benefits of group-living outweigh overall costs [6,7]. In ground-dwelling sciurids, the expression of post-weaning cohesive behaviours between adults and young has been regarded as a key factor promoting the evolution of sociality [9,10]. The expression of predominantly cohesive behaviour (i.e. increased affiliation and reduced agonism), along with delayed dispersal and reproduction are argued as necessary in the face of short active seasons, which would otherwise lead to increased mortality [9,11]. In ground squirrels, non-dispersing individuals are probably offspring or other maternal kin and thus kin selection may further promote these social groupings [9,12]. However, selection may ultimately act to broaden the expression of beneficent behaviour among neighbouring conspecifics to encompass kin and non-kin alike [13].

## 1.1. Changes in the family unit

Research on parental removal has focused primarily on removing the father to explore the contributions of males in the context of biparental care. Removal of a parent can influence the hypothalamic-pituitary-adrenal (HPA) axis, resulting in altered cortisol or glucocorticoid production, [14,15] and these changes can have lasting effects on offspring behaviour [16]. The few studies addressing the effects of mother removal during the post-weaning period have yielded inconsistent results in terms of the effects on offspring. In bushy-tailed woodrats (*Neotoma cinerea*), the presence of a mother in the natal area improved the survival of her female young, but only when overall densities of woodrats were high [17]. In red deer (*Cervus elaphus*), young orphaned in the post-weaning period had lower survival [18]. In grey-sided voles (*Myodes rufocanus*), field studies of mothers placed in food-supplemented enclosures revealed that offspring did not differ in their space use, dispersal or social behaviour when mothers were removed in the post-weaning period [19]. Thus, previous research has revealed a variety of physiological, behavioural and fitness impacts on offspring deprived of parental care, yet sometimes, no effect at all.

## 1.2. Changes in the social environment

In addition to the impacts of changing parental care, changing an animal's broader social environment can influence survival, reproductive fitness and physiology. The direction of these effects are largely species and life-history specific and are influenced by myriad external and internal factors (e.g. population density, breeding state). For example, in marmots, being in close kin groups improved survival (*Marmota marmota* [20]) and reproductive success (*Marmota flaviventris* [21]), while only reproductive success was improved in field voles (*Microtus agrestis*) and mice (*Mus domesticus*) [22,23]. However, no effect on survival or reproductive success was detected by Boonstra & Hogg [24] for meadow voles (*Microtus pennsylvanicus*), or by Dalton [25] for grey-tailed voles (*Microtus canicaudus*), and in some cases, the presence of kin can prove detrimental in terms of reproductive success or

offspring survival as reported by Hoogland [26] for black-tailed prairie dogs (*Cynomys ludovicianus*) and Lacey [5] for social tuco-tucos (*Ctenomys sociabilis*).

In studies where the social environment has been manipulated, removal of individuals typically increases immigration by other individuals from neighbouring areas (Clutton-Brock & Lukas [27]), which can improve offspring survival if the offspring's mother acquires a nearby territory [28]. In social species such as the Alpine marmot (*Marmota marmota*), and the degu (*Octodon degus*), changes in the social environment can impact reproductive success by altering intrasexual competition, and opportunities for cooperative offspring care, respectively [29,30]. Altering the social landscape can also impact the HPA axis for those in a population, as changes in social relationships, territories and dominance hierarchies can increase circulating glucocorticoids, which could also alter fitness [31,32].

## 1.3. Richardson's ground squirrels

Richardson's ground squirrels (*Urocitellus richardsonii*) are a social, colonial rodent that lives in matrilineal kin groups [33,34]. This species has a female philopatry-based colonial system where males disperse, and females typically remain on or adjacent to their mother's core area. This presence of related individuals influences how individuals respond to predators—in the presence of kin, females are more likely to alarm call and potentially draw attention to themselves [34]. In doing so, however, these alarm callers allow kin to escape, which can result in indirect fitness benefits [35]. In addition to this nepotistic behaviour, squirrels are able to differentiate between neighbours and non-neighbours and respond more strongly to alarm calls from familiar individuals [33,36], but not their own mother [37]. After emerging from hibernation in the spring but before giving birth to young, adult female kin will often share sleeping burrows [38], though there is no evidence that they nest with or care for (i.e. alloparent or allonurse) the pre-weaned offspring of kin. Throughout the season, kin will overlap in area use, and adult females are more tolerant of close relatives [39,40]. In the population we studied, juvenile ground squirrels emerge from their natal burrow in late May and early June. It is clear that prior to emergence females provide essential care to their young through gestation, lactation and associated maternal care. After emergence, females strongly defend core areas from intruding females [39] and engage in increased alarm-calling behaviour [34]. At the time of emergence, predation of offspring is common; if mothers breed early they provide additional time for offspring to acquire mass for hibernation [41]. However, by synchronizing their breeding efforts with other females mothers can theoretically reduce the risk of predation of their own offspring by saturating predator appetites and capitalizing on others' alarm calling. In this species, adult females engage in a variety of post-weaning behaviours that are important for offspring care. We do not know how the presence of mothers in this post-weaning period influences juvenile and adult kin. Given the important roles that relatedness, familiarity and sociality play in this species, we hypothesized that these factors might also influence an individual's perceived stress level as indicated by their physiological stress response and subsequent fitness. We seized the opportunity to study the impact of adult female removals, which constituted an essential component of a contemporaneous study [42] on both the removed females' offspring and on neighbouring conspecifics by quantifying immediate stress axis activation via faecal glucocorticoid metabolites, overwinter survival and reproductive success of those individuals in the year subsequent to the removals.

## 1.4. Hypotheses

We hypothesized that changing the social environment of Richardson's ground squirrels would influence the stress response of conspecifics and alter future survival and reproduction (i.e. reproductive timing and success). We predicted that changing the social environment would increase faecal glucocorticoid metabolites and result in reduced survival and altered reproduction of conspecifics.

We tested the following *a priori* hypotheses:

H1. Removal of a mother would alter glucocorticoids of offspring, and alter these differently depending on age and sex of the offspring

H2. Removal of a mother would influence the survival and reproduction of offspring

H3. Removal of a neighbour would alter glucocorticoids in remaining neighbours and have differential effects if that neighbour was kin.

H4. Removal of a neighbour would influence the survival and reproduction in remaining neighbours, and have differential effects if that neighbour was kin.

We made the following predictions for offspring: removal of the mother would increase faecal glucocorticoid metabolites in her weaned offspring due to increased stress from the reduction of maternal care, and these effects would be influenced by age and sex-specific dispersal by offspring (H1). We predicted that this increase in glucocorticoids would lead to reduced overwinter survival and impaired reproduction in female offspring (H2).

We made similar predictions for neighbours: if individuals are obtaining benefits from their neighbours (e.g. through social stability, alarm calling, kinship, reciprocity or by-product mutualism), we should see increased faecal glucocorticoid metabolites (H3) and reduced survival and reproduction of neighbours when others are removed (H4). We further predicted that the impacts on kin would be of a greater magnitude compared to non-kin, since kinship plays an important role in other behaviours (e.g. alarm calling) in this species [34] (H3 and H4).

# 2. Methods

## 2.1. Population, trapping and female removal

We conducted research in 2014, 2015 and 2016 at the Assiniboine Park Zoo in Winnipeg, Manitoba (49°52′ N, 97°14′ W). This free-living population of Richardson's ground squirrels located in the former picnic grounds has a record of matrilineal relatedness for all squirrels born within the population from 2004 onward and is habituated to human presence. We trapped squirrels using Tomahawk live traps (Tomahawk Live Trap Co., Tomahawk, WI, USA) baited with peanut butter, permanently marking squirrels with numbered metal ear tags (National Band and Tag Co. Monel #1, Newport, KY, USA), and with distinctive marks applied to the dorsal pelage with human hair dye (Clairol Hydrience, Black Pearl, Stamford, CT, USA). We trapped females and males during the breeding season (March and April) and noted copulatory plugs or caked semen around the vaginal area of females to record breeding dates and estimate parturition dates [43]. From these observations, we estimated when juveniles for each mated female were likely to emerge and closely observed female nest burrow entrances to confirm juvenile emergence and to trap juveniles. We trapped juveniles as they emerged from their natal burrows, marking individuals as described above. All juveniles in this study were from litters for which the mother was identified. Individuals from a single litter were assigned a coefficient of relatedness of 0.5, given we had only maternal relatedness data from our pedigree records, and because juveniles behaviourally discriminate littermates from non-littermates, even though some individuals were undoubtedly maternal half-siblings due to multiple paternity [44,45]. Neighbours were defined as individuals that had core areas adjacent to one another and nest burrows that were less than 30 m apart. We defined 'neighbourhoods' as larger clusters of neighbours within the colony which were separated by physical features of the landscape. All squirrels in the population were trapped at least once each summer.

From 16 June to 28 July 2014, and 8 to 29 June 2015, we either removed (euthanized in the context of a separate experiment $n = 22$) or observed the predation of ($n = 4$) adult females with known litters, after their juveniles had emerged from the natal burrow. All trapping, marking, handling and euthanization of animals was conducted in accordance with the guidelines set forth by the Canadian Council on Animal Care, approved under protocols F13-014/1 of the University of Manitoba Fort Garry Campus Animal Care Committee, 2014-A003 of the Assiniboine Park Zoo Research Ethics and Review Committee, and Wildlife Scientific Permit WB14952 from Manitoba Conservation.

## 2.2. Experimental faecal samples

We collected faecal samples from offspring and adult female neighbours (hereafter, 'neighbours') in territories adjacent to females scheduled for removal prior to female removal (2014: range: 0 (same day)–37 days, mean: 10.5 days; 2015: range: 0–10, mean: 3.7 days), and approximately 4 days after female removal (2014: range: 1–5 days, mean: 3.8 days, 2015: range: 2–12 days, mean: 4.8 days). While we attempted to collect samples 4 days after mother removal, and have a relatively recent paired sample, uncontrollable circumstances in the field including weather and predation precluded the collection of all samples on day 4. Faecal glucocorticoid metabolites peak 3–5 days after a stressful event, as recorded in a previous study validating our faecal glucocorticoid metabolite assay [46]. To account for this, we analysed a conservative dataset including only values within the 3–5-day window and compared these results with those obtained using the complete dataset. All faecal samples were

collected from 8.00 to 15.00 CDT, with variation in timing due to above-ground activity, and willingness of individuals squirrels to enter live-traps. We collected fresh faeces within 30 min of defecation using clean, disposable wooden sticks. Only faeces uncontaminated with urine were collected.

## 2.3. Control faecal samples

Control faecal samples were collected from neighbours and offspring of adult females that were not euthanized and that were observed alive in late July prior to hibernation. We collected *ad hoc* faecal samples from neighbours and offspring in the population throughout the summer, sampling from neighbours adjacent to where females were being removed. Once we confirmed which females had survived late into the season, we randomly selected a female in the same neighbourhood as the associated control. Control samples were matched to removed females in the same area by collecting samples in the same week as matched experimental samples.

## 2.4. Faecal sample analysis

All faecal samples were placed in plastic scintillation vials and stored at −20°C until analysis. Up to two kin and two non-kin neighbours were assayed for each removed or control adult female, and one juvenile offspring of each sex (where available) were assayed and analysed. In some cases, a neighbour could serve as a data point as 'non-kin' for one individual and 'kin' for another at different times in the study. Since female removals occurred continually through the season, we collected multiple samples from neighbours which served as multiple datapoints. While we attempted to avoid re-selecting neighbours for faecal sampling, in some instances (2014: 9 of 30; 2015: 13 of 43) the only available 'kin' neighbours were the only available 'non-kin' neighbours for others.

We analysed faecal samples for glucocorticoid metabolites using methods described in Ryan *et al.* [47], and validated in Richardson's ground squirrels by Hare *et al.* [46], using a 1 : 9000 cortisol-specific antibody dilution [46,47]. Our intra-assay coefficient of variation in 2014 was 12.4%, while the inter-assay coefficient of variation was $13.4 \pm 6.6\%$. In 2015, the intra-assay coefficient of variation was 10.9%, while the inter-assay coefficient of variation was $14.6 \pm 4.6\%$. All controls were assayed in triplicate, while samples were assayed in duplicate. We rejected aberrant samples that had standard deviations of greater than 125 dpms (disintegrations per minute) and re-assayed those samples in duplicate or triplicate. The treatment group of each sample (removed or control) was blinded to experimenters conducting the assays.

## 2.5. Survival and reproduction

Survival and reproductive measures were collected after hibernation (in the following year) once for each responding individual (i.e. female offspring or female neighbour). Each adult female's breeding date and offspring were recorded during yearly live-trapping, as noted above. Males were excluded from all reproductive and survival analyses (unless noted) due to their dispersal outside of the study population. Since neighbours could have multiple removals of kin or non-kin within their neighbourhood, we used the total proportion of neighbours removed, and whether those removed were kin, as the predictors of survival or reproduction for neighbours (table 1). Breeding date shift was calculated as the date of breeding for the individual subtracted from the mean breeding date within the population for that year.

## 2.6. Data analysis

We used R v. 3.3.2 and the lme4 package to fit linear mixed models (LMMs) and generalized LMMs (GLMMs) to the data [48,49]. In all models, neighbourhood identity and the identity of the removed/control female were included as nested random effects.

To test for any effect of female removal on offspring and neighbour glucocorticoids (H1, H3), we ran a LMM using faecal glucocorticoid metabolite change scores (difference in faecal cortisol, in ng g$^{-1}$ faeces) for each individual as the response variable. For offspring (H1), year (2014 or 2015) and treatment group (removed or control) were used as fixed effects (with interaction). For neighbours, year, treatment group and relatedness were included as explanatory variables. We included year as a fixed effect in our analyses to help account for differences in predator abundance, food availability and weather between years. Relatedness for neighbours was calculated as the coefficient of relatedness (*r*) between individuals [50]. Offspring and neighbours were analysed in separate models since

**Table 1.** Summary of sampled individuals by grouping.

| | year | focal females | kin neighbour | non-kin neighbour | male offspring | female offspring |
|---|---|---|---|---|---|---|
| removal | 2014 | 10 | 7 | 17 | 6 | 8 |
| control | 2014 | 9 | 10 | 13 | 4 | 4 |
| removal | 2015 | 16 | 21 | 31 | 12 | 12 |
| control | 2015 | 11 | 16 | 22 | 11 | 10 |

offspring did not vary in relatedness to their mother. To test for age and sex effects on faecal glucocorticoid metabolite change scores in offspring (H1), we used a LMM with treatment group, offspring age, offspring sex, all interactions and year as fixed effects. For each of the above models, we analysed both the full dataset, and a subset which included all controls, but only included treatment samples which were collected 3–5 days after female removal, since faecal glucocorticoid metabolites peak 3–5 days after a stressful event [46]. Additionally, we examined how offspring faecal glucocorticoid metabolites (not change scores) were predicted by age using a linear model.

To analyse offspring survival (H2), we used a binomial GLMM with a log link function to model whether or not offspring survived. Fixed effects included treatment (mother removal or control) and year. To analyse neighbour survival (H4), we used a similar binomial GLMM with the proportion of neighbours removed, whether kin were removed (and interactions) and year (without interaction) as fixed effects.

To analyse neighbour reproductive success (H4), we used a Poisson GLMM with a log link function, with the number of offspring produced (at emergence) as the response variable. The proportion of neighbours removed, whether kin were removed, their interactions and year were included as fixed effects. We also analysed the timing of neighbour reproduction by examining their relative breeding dates. We calculated each neighbour's deviation from the mean breeding date for each year (in days). This value was used as the response variable in an LMM which included the proportion of neighbours removed, whether kin were removed, and the year (including interactions) as fixed effects.

Conditional $R^2$ is reported as an effect size for LMM and GLMMs using the MuMIn package [51,52]. Binomial and Poisson models were tested for overdispersion using the $\chi^2$ method for the observed and predicted variance [53]. The 95% confidence intervals for LMM and GLMMs were estimated using the Wald method. *Post hoc* analyses of neighbour breeding date using the Tukey correction were conducted with the emmeans package [54].

## 3. Results

In 2014, we removed 10 females and obtained parallel data for nine surviving control females. In 2015, we removed 16 females and obtained data for 11 control females (table 2). On the study site in 2014, 2015 and 2016 there were 111, 86 and 66 marked adult females and 349, 325 and 317 marked offspring, respectively. We collected a total of 414 faecal samples from 143 individual squirrels.

### 3.1. Impacts on offspring

> H1. Removal of a mother would alter glucocorticoids of offspring differently depending on age and sex of the offspring

Neither year, removal of the mother, nor their interaction predicted the faecal glucocorticoid metabolite change in offspring (LMM, conditional $R^2 = 0.13$, mean ng g$^{-1}$ ± s.d.: removed ($n = 38$): $-0.50 \pm 2.9$, control ($n = 29$): $-0.10 \pm 2.2$; table 3). When considering experimental group samples collected in the 3- to 5-day window after mother removal (figure 1), neither removal of the mother, year, nor their interactions predicted the faecal glucocorticoid metabolite change in offspring (LMM, conditional $R^2 = 0.06$, mean ng g$^{-1}$ ± s.d.: removed ($n = 9$): $-0.74 \pm 1.9$, control ($n = 29$): $-0.10 \pm 2.2$; electronic supplementary material, table S1).

Offspring sex, age, and whether its mother was removed did not predict (separately or in any interaction) the change in faecal glucocorticoid metabolite values (LMM, conditional $R^2 = 0.19$, mean ng g$^{-1}$ ± s.d.: removed group males ($n = 18$): $-0.64 \pm 2.19$, control group males ($n = 15$): $-0.14 \pm 1.85$, removed group females ($n = 20$): $-0.37 \pm 3.47$, control group females ($n = 14$): $-0.06 \pm 2.59$; electronic

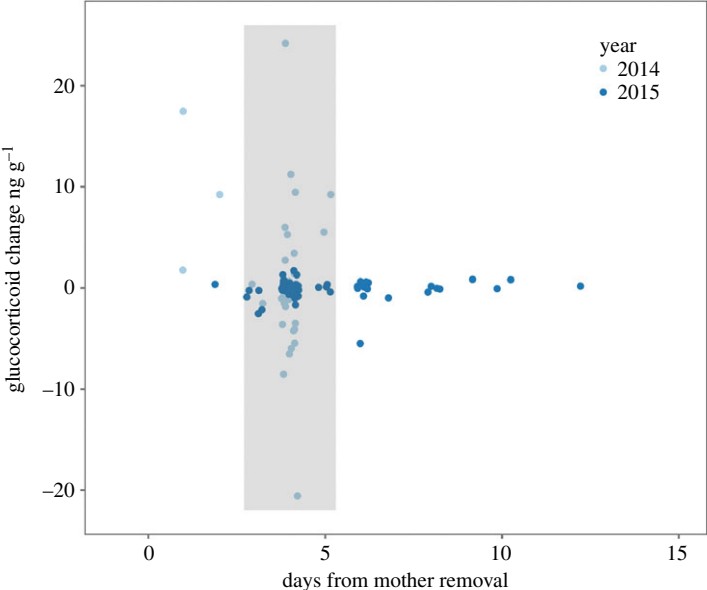

**Figure 1.** Scatterplot of faecal glucocorticoid metabolite change values for juveniles within the removal group. *X*-axis indicates timing of sampling from mother removal. Light blue points are data from 2014, dark blue points are data from 2015. The grey box indicates the 'ideal window' for glucocorticoid metabolite change associated with a stressful event.

**Table 2.** Number of individuals sampled with kin neighbours removed (greater than or equal to one kin removed) or only non-kin neighbours removed, by the number of neighbours removed for that individual.

|  | no neighbours removed (control) | one neighbour removed | two neighbours removed | three neighbours removed |
|---|---|---|---|---|
| kin neighbour(s) removed | n.a. | 17 | 4 | 3 |
| only non-kin neighbour(s) removed | n.a. | 22 | 4 | 4 |
| total number | 18 | 39 | 8 | 7 |

**Table 3.** Linear mixed model summary of faecal glucocorticoid metabolite changes in offspring following either the removal of the mother or no removal (control) ($n = 67$).

|  | estimate | s.e. | 95% CI lower | upper | $t$ | *p*-value |
|---|---|---|---|---|---|---|
| intercept (ref: 2014, no removal) | 0.308 | 0.99 | −1.64 | 2.25 | 0.31 | 0.76 |
| mother removed | −0.907 | 1.24 | −3.34 | 1.53 | −0.73 | 0.47 |
| year (2015) | −0.583 | 1.17 | −2.87 | 1.70 | −0.50 | 0.62 |
| removal : year | 0.770 | 1.50 | −2.16 | 3.70 | 0.52 | 0.61 |

supplementary material, table S2). Including only experimental samples collected 3–5 days post mother removal did not result in rejection of the null (electronic supplementary material, table S3). Older juveniles tended to have increased faecal glucocorticoid metabolite values (LM, $F_{1,62} = 7.05$, $p = 0.01$, adj-$R^2 = 0.09$, $\eta^2 = 0.102$), though the overall effect size was small.

H2. Mother removal would influence the survival and reproduction of offspring

Dispersal of male offspring prevented an accurate assessment of male survival to the following year; however, two males in 2014 that had dispersed within the study population area were recaptured in 2015

**Table 4.** Distribution of offspring survival by treatment.

|         | survived               | did not survive | unknown (males) |
|---------|------------------------|-----------------|-----------------|
| removal | 7                      | 14              | 17              |
| control | 3 (includes 2 males)   | 12              | 14              |

**Table 5.** Linear mixed model summary of faecal glucocorticoid metabolite changes in neighbours following kin or non-kin neighbour removal or no neighbour removal (control) ($n = 140$).

|                                              | estimate | s.e. | 95% CI lower | 95% CI upper | $t$    | $p$-value |
|----------------------------------------------|----------|------|--------------|--------------|--------|-----------|
| intercept (ref: 2014, no removal, 0 relatedness) | 1.72     | 1.35 | −0.93        | 4.37         | 1.27   | 0.21      |
| neighbour removed                            | 0.60     | 1.74 | −2.81        | 4.01         | 0.35   | 0.73      |
| relatedness coefficient                      | −0.61    | 4.74 | −9.90        | 8.68         | −0.13  | 0.90      |
| year (2015)                                  | −0.53    | 1.65 | −3.77        | 2.71         | −0.32  | 0.75      |
| removal : relatedness                        | −2.62    | 6.92 | −16.2        | 10.94        | −0.38  | 0.71      |
| removal : year (2015)                        | −1.66    | 2.24 | −6.05        | 2.72         | −0.74  | 0.46      |
| relatedness : year (2015)                    | −0.88    | 6.07 | −12.78       | 11.03        | −0.14  | 0.89      |
| removal : relatedness : year                 | 4.81     | 8.56 | −11.97       | 21.59        | 0.56   | 0.58      |

(table 4). Including these recaptured males, we did not detect any significant influence of mother removal or year on the likelihood of survival of offspring (GLMM, delta conditional $R^2 = 0.03$, $n = 39$, removal: $z = 0.1$, $p = 0.92$, year: $z = −0.09$, $p = 0.93$, interaction: $z = 0.47$, $p = 0.64$; electronic supplementary material, table S4). Of the eight surviving female offspring, all were successful breeders in the following year, and had similar numbers of pups (pup means ± s.e.: removal: $4.4 ± 1.4$, control: $7.5 ± 2.0$). A small number of surviving offspring prevented additional analyses of reproduction.

## 3.2. Impacts on neighbouring females

> H3. Removal of a neighbour would alter glucocorticoids in remaining neighbours and have differential effects if that neighbour was kin

The removal of a neighbour, the relatedness of that neighbour, and the year did not predict faecal glucocorticoid metabolite changes in neighbours (LMM, conditional $R^2 = 0.11$, mean ng g$^{-1}$ ± s.d., removed neighbour ($n = 77$): $0.61 ± 4.85$, control neighbour ($n = 63$): $1.00 ± 5.56$; table 5). Revising the model to only include experimental samples which were taken 3–5 days after female removal did not lead to rejection of the null (LMM, conditional $R^2 = 0.13$, mean ng g$^{-1}$ ± s.d., removed neighbour ($n = 20$): $0.70 ± 3.08$, control neighbour ($n = 63$): $1.00 ± 5.56$; electronic supplementary material, table S5).

> H4. Removal of a neighbour would influence the survival and reproduction in remaining neighbours, and have differential effects if that neighbour was kin

We did not detect any impact of removing of closely related neighbours on surviving neighbours. The proportion of removed neighbours and whether those neighbours were kin (table 1) did not influence survival of adult females (GLMM, delta conditional $R^2 = 0.14$, survivors when no kin removed: 11/48, survivors when kin removed: 15/24; electronic supplementary material, table S6). The number of offspring produced in the following year by surviving neighbour females was not influenced by the proportion of neighbouring females removed, the year, or whether the removed females were kin (table 6).

For 2015, the mean breeding date for the population ($n = 59$ with known breeding dates) was 6 April 2015 (median: 6 April 2015, mode: 7 April 2015, range: 26 March–21 April 2015). For 2016, the mean breeding date for the population ($n = 62$ females with known breeding dates) was 29 March 2016 (median: 27 March 2016, mode: 26 March 2016, range: 20 March–18 April 2016).

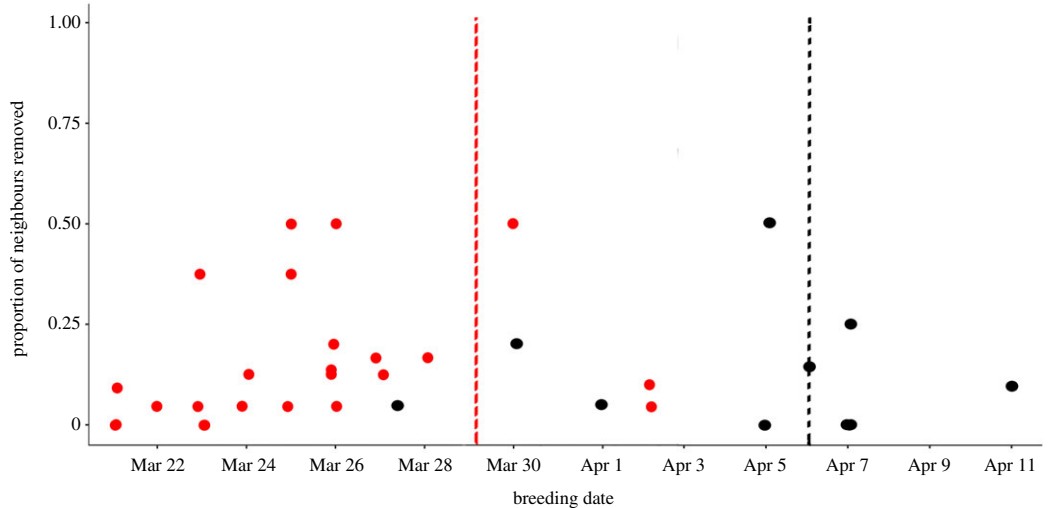

**Figure 2.** Scatterplot of neighbour breeding date and the proportion of neighbours removed. Breeding dates are from year subsequent to removals. Mean breeding date for 2015 indicated with black dashed vertical line. Mean breeding date for 2016 indicated with red dashed vertical line. Black points represent individuals from the 2014 season, red points represent individuals from the 2015 season. The proportion of neighbours removed did not influence breeding date in either year.

**Table 6.** Poisson GLMM summary of the number of offspring produced by female neighbours following kin or non-kin neighbour removal, or no neighbour removal (control) ($n = 37$).

| | estimate | s.e. | 95% CI | | z | *p*-value |
| | | | lower | upper | | |
|---|---|---|---|---|---|---|
| intercept (ref: 2014, 0 proportion removed, no kin removed) | 1.816 | 0.19 | −1.27 | 0.76 | 9.64 | <0.001 |
| proportion neighbours removed | −0.848 | 0.75 | −3.84 | 1.44 | −1.13 | 0.26 |
| kin removal (binary; Y) | 0.224 | 0.23 | −0.62 | 2.61 | 0.97 | 0.33 |
| year (2015) | −0.085 | 0.20 | −0.35 | 1.93 | −0.43 | 0.67 |
| proportion removed : kin removal | 0.501 | 1.03 | −6.92 | 3.74 | 0.49 | 0.63 |

We were able to determine the breeding date of all surviving neighbouring females in 2015, though our small sample size was due to low survival rates after hibernation. In 2016, we were unable to determine the breeding date for one individual who had non-kin neighbours removed. We detected no effect of the proportion of neighbours removed on breeding date shift, and year did not interact with removal to predict breeding date shift (figure 2 and table 7). However, there was a significant effect of year and kin removal on breeding date (table 7). In 2014, when kin were removed an individual's breeding date shifted significantly (*post hoc* $t_{27} = 3.478$, estimate = 7.82, s.e. = 2.25, $p = 0.002$) in the following year, with the mean ± s.d. date shift in 2015 when at least one kin neighbour was removed being −5.75 ± 3.8 days ($n = 4$) compared to 1.14 ± 1.9 days later ($n = 7$) when no kin were removed. We detected no impact of the removal of kin on breeding date for animals studied in 2015 in the following year (*post hoc* $t_{18} = -1.056$, estimate = −1.33, s.e. = 1.26, $p = 0.30$).

## 4. Discussion

Our data reveal that Richardson's ground squirrels are particularly resilient to the removal of their mother after weaning or to the disappearance of a neighbour, and do not show any change in faecal glucocorticoid metabolites immediately after removal. Furthermore, we did not detect any changes in

**Table 7.** Linear mixed model summary of breeding date shift for neighbours following kin or non-kin neighbour removal, or no neighbour removal (control) ($n = 35$).

| | estimate | s.e. | 95% CI lower | 95% CI upper | t | p-value |
|---|---|---|---|---|---|---|
| intercept (ref: 2014, 0 proportion removed, no kin removed) | 0.94 | 1.56 | −2.12 | 4.00 | 0.61 | 0.55 |
| proportion neighbours removed | 2.47 | 13.08 | −23.17 | 28.10 | 0.19 | 0.85 |
| kin removal (binary; Y) | −9.62 | 2.73 | −15.00 | −4.26 | −3.52 | 0.002 |
| year (2015) | −5.57 | 1.93 | −9.35 | −1.79 | −2.89 | 0.009 |
| proportion removed : kin removed | 12.52 | 13.9 | −14.7 | 39.7 | 0.90 | 0.38 |
| proportion removed : year (2015) | −0.22 | 13.9 | −27.4 | 27.0 | −0.02 | 0.99 |
| kin removed : year (2015) | 10.34 | 3.23 | 4.01 | 16.7 | 3.20 | 0.004 |
| Prop. removed : kin removed : year | −8.27 | 15.8 | −39.2 | 22.7 | −0.52 | 0.61 |

offspring or neighbour survival, or in neighbour reproductive success after female removal. We had originally predicted that the removal of an individual from the population would influence the release of glucocorticoids among surviving conspecifics to prepare for any change in territory size (i.e. changes in the number of agonistic encounters), associated resource competition, or predation risk (fewer alarm callers). We further predicted that these changes would result in altered survival and reproduction due to shifting energetic costs and predation risk. Richardson's ground squirrels are either not influenced significantly by these social environment changes, are influenced over the longer term, or perhaps the costs and benefits of these changes negate each other resulting in no effective influence. In any case, the resilience of these individuals in the post-weaning period may be a response to the high levels of mortality that exist within ground squirrel populations and serve as an adaptation to their variable social environment.

## 4.1. The stress response

Given that juveniles in this study showed no changes in faecal glucocorticoid metabolites immediately after their mother was removed, it appears that post-emergence juvenile Richardson's ground squirrels are self-sufficient. Indeed, by the end of July, most adults have entered hibernation, leaving the juveniles on their own to gain the mass necessary to survive the winter [12]. Potentially, the impacts of mother removal were more nuanced or might be more apparent in the long-term, as prenatal, infant or juvenile, and adult stress have different impacts on the brain and behaviour [55]. In work investigating social stress on juvenile rats, chronic social stress altered the neuroanatomy of the hippocampus (soma size) as well as changing glucocorticoid receptor mRNA expression in the short term [56].

In animals where biparental care is typical, parental absence is associated with reduced offspring survival [57–59], growth [60] and immunocompetence [60], and increased behavioural stereotypy [61], among other behavioural changes (reviewed in [62]). Some of these effects of parental loss, however, can be offset in resource-rich habitats [63,64]. In addition to impacts on survival, the removal of a parent can influence the stress response in offspring leading to changes in serum cortisol or glucocorticoids, or their associated receptors [15,65–67]. These parental-deprivation studies are typically conducted prior to weaning or fledging, during the extremely energetically costly lactation or nestling period.

Unfortunately, most studies examining the effects of social stress on juveniles have focused on the effects of complete isolation on physiology and behaviour [68], creating an all-or-none design which emphasizes the impacts of social stress [69,70]. For social rodents, social stress is rarely all-or-none and instead encompasses a gradient of stable to unstable social environments comprised of animals with a wide range of social familiarity or relatedness. The removal of only a few individuals in this population had no detectable effect on survivors, which suggests that this level of instability is insufficient to significantly impact individuals. Richardson's ground squirrel individuals retain

memories of past interactions with known individual conspecifics within their social environment [71]; thus, variance in the nature of these relationships might influence the impact of removal on those individuals. While Richardson's ground squirrels are social, live in large colonies, and benefit from shared defence and vigilance towards predators, they do not alloparent the offspring of kin, and males do not engage in paternal care. Whether similar species which alloparent or are biparental have similar resilience to social change remains an open question; though evidence from two highly social species suggests that changes in the social environment might detrimentally impact the number of weaned offspring [30] and their survival [29].

We predicted that the initial removal of the mother would be a 'stressful' stimulus for offspring and attempted to collect and compare samples derived from the critical window of 3–5 days after the stressor. However, including samples which were further outside this 'ideal window' did not influence the significance in our model or our interpretation. Thus, we concluded that variance in sampling dates did not account for the failure to detect any measurable effect of female removal. We did not assess short-term (i.e. within minutes to hours) changes in glucocorticoids which may have revealed any acute stress response. Furthermore, we cannot assess whether certain individuals may have been exposed to other stressful events which we did not observe. Because squirrels are able to move freely among burrows and around the study site, each individual will have unique experiences. We have controlled for neighbourhood edge- and individual-effects but cannot exclude the possibility that differences in above-ground activity could result in different experiences for different individuals.

The absence of kin-related effects on faecal glucocorticoid metabolites suggests that these changes in the social environment, which we postulated would act as 'social stressors', are not perceived by individuals, not perceived differently by kin and non-kin, or perhaps, might be buffered by the presence of the remaining individuals [72]. Furthermore, Bairos-Novak *et al.* [73] recently demonstrated that variation in the cortisol-based stress response of Richardson's ground squirrels is attributable more to heritability than environmental influences, suggesting that their physiological stress response may be more robust to external stressors than previously assumed. We attempted to collect faecal samples for glucocorticoid metabolite analysis from a variety of neighbours, though we observed that individuals vary in their 'trapability'. This natural variation might correlate with glucocorticoid physiology. We made a concerted effort to collect samples from both 'trap-shy' and 'trap-bold' individuals to encompass the natural range of phenotypes. Whether trap aversion is predicted by faecal glucocorticoid metabolites in Richardson's ground squirrels is currently unknown. Additional investigation of social buffering and glucocorticoids, as well as investigation of glucocorticoid receptors, is necessary to determine if removal of a mother or neighbour influences the stress response in more subtle ways including effects over multiple years and generations.

## 4.2. Survival to adulthood

We had predicted that changing the social environment during a critical period (i.e. adolescence) would influence survival to adulthood. However, we detected no differences in survival for young whose mothers were removed compared to controls. Survival to adulthood is very low for juvenile Richardson's ground squirrels relative to other sciurids; estimates from other populations suggest 2–30% of male and 30–55% of female juveniles survive to become breeding yearlings [12,74]. If mothers are giving essential post-weaning care, in the form of territorial and predator defence (i.e. alarm calling), we would expect to see a difference in the survivorship of offspring with and without mothers. In a field study of grey-sided voles, young whose mothers were removed post-weaning did not differ in space-use, dispersal, or social behaviour, though long-term effects such as overwinter survival and reproductive success were not assessed [19]. For bushy-tailed woodrats, survival was improved for juveniles when the mother was present, but only when resources were limited, and there were no significant maternal effects on reproductive success [17]. In other species, adult removal within a population can improve juvenile survival [75,76], or at least does not hinder juvenile survival [77]. It is possible that female juveniles whose mothers are removed experience conflicting positive and negative effects (e.g. reduced competition and increased predation risk) that result in no net change in survival. Furthermore, we cannot exclude the possibility that male juveniles may have experienced differential survival after mother removal, though due to dispersal to areas outside of the study population, we cannot accurately assess the survival of male juveniles.

We have postulated that resilience to mother loss is adaptive in Richardson's ground squirrels since predation in this species is relatively high. In species where predation is less common, removal of a mother in the post-weaning period might be perceived differently and impact survival—however,

there are few studies which have directly examined this hypothesis. For red deer, survival rates of young are fairly high (approx. 67 to 73% depending on population; [78,79]), and loss of a mother post-weaning is detrimental to offspring survival [18]. However, further comparative data are necessary to determine if these effects on survival are seen among other species where juvenile mortality is relatively low.

## 4.3. Effects on offspring reproduction in adulthood

The presence of a mother did not have any measurable impact on subsequent reproduction among offspring. We had hypothesized that kin might provide social facilitation which could lead to improved reproductive success [80,81], though alternatively, mother–daughter conflict could result in non-breeding yearlings [81,82]. We observed no trend in either direction—reproductive success in surviving female offspring was not influenced by the presence or absence of their mother. Given that the sample size of surviving offspring was predictably low, we may not have had the statistical power to resolve small differences in reproductive performance, or the effects themselves may only be realized over the life of the organism. In either case, no effects on reproductive success were apparent in the year after female removal. All female offspring were successful at breeding (as determined by the presence of copulatory plugs or other evidence of breeding), and all but two females successfully reared young to emergence.

## 4.4. Kinship

Given the nepotistic alarm calling of Richardson's ground squirrels [34], we predicted that removal of a neighbour might differentially affect kin and non-kin. Faecal glucocorticoid metabolites immediately after female removal did not vary between kin, non-kin and controls. Furthermore, the number of removed neighbours, the proportion of removed neighbours and whether those neighbours were kin did not have a detectable influence on reproductive success or survival of neighbours. We noted an effect of the 2014 kin neighbour removal on the 2015 breeding date of surviving females. However, the shift in breeding date did not significantly affect reproductive output as females successfully raised similar numbers of pups to emergence independent of the number of neighbours removed. In Richardson's ground squirrels, breeding earlier can prove advantageous—as young have additional time to gain mass prior to hibernation [41]. However, due to overwinter death or dispersal of offspring discussed previously, we probably did not have a large enough sample of yearlings in 2015 from these neighbours to determine whether the shift in breeding date had any measurable effect on offspring. Additionally, the change in breeding date was not evident in the following year's cohort of neighbours, suggesting that year effects may also play a role. Thus, we did not detect any significant influence of kinship on the effect of neighbour removal for Richardson's ground squirrels, consistent with the failure to detect any differences in the frequency of cohesive, agonistic or recognitive behaviours between closely related versus unrelated pairs of neighbouring juvenile Richardson's ground squirrels in an earlier study exploring social discrimination [44].

## 5. Summary

Richardson's ground squirrel juveniles are resilient to changes in their social environment after emergence, including the loss of their mother. For these small mammals, predation, dispersal, immigration and overwinter mortality continually alter their social landscape. Via an unknown mechanism, which could include social buffering, or perhaps trade-offs between competing costs and benefits, juveniles who had their mothers removed had similar rates of survival to those whose mother survived to hibernation. Similarly, neighbours of these removed females were also resilient, showing no changes in survival or reproduction relative to changes in their social environment, including the removal of kin neighbours. Thus, we conclude that Richardson's ground squirrels may have evolved resilience to the changing social landscape and are, at least as far as our results can resolve, largely unaffected by the loss of a close relative.

Research ethics. All applicable international, national and/or institutional guidelines for the care and use of animals were followed.
Animal ethics. All procedures performed in studies involving animals were in accordance with the ethical standards of the Canadian Council on Animal Care, the University of Manitoba Fort Garry Campus Animal Care Committee

(Protocol no. F13-014/1), the Assiniboine Park Zoo Research Ethics and Review Committee (Protocol no. 2014-A003) and Manitoba Conservation (Wildlife Scientific Permit WB14952).

Permission to carry out fieldwork. Collection and use of ground squirrels were approved by Manitoba Conservation (Wildlife Scientific Permit WB14952).

Data accessibility. Underlying data and code are available through the open science framework at: https://osf.io/apv3r/?view_only=1fc5739c407d4af2a4d91b0a46b39f1b [83].

Authors' contributions. A.R.F. and J.F.H. conceived of and designed the study. A.R.F., T.J.W., K.R.B.-N. and J.F.H. conducted fieldwork. A.R.F., T.J.W. and K.R.B.-N. conducted laboratory analyses. W.G.A. designed, supervised and coordinated all laboratory analyses. A.R.F. and K.R.B.-N. conducted data preparation and analysis. All authors interpreted data, helped draft the manuscript and gave final approval for publication.

Competing interests. The authors declare they have no conflict of interest.

Funding. Funding from the Natural Sciences and Engineering Research Council of Canada (NSERC) supported this work; Discovery grant funds awarded to J.F.H. (#154271) and W.G.A. (#311909) and a NSERC postgraduate scholarship to A.R.F. (#PGSD3-426556-2012). J.F.H. was further supported by a University of Manitoba Field Work Support Program Grant.

Acknowledgements. The authors thank the following individuals for their assistance: field crew: Justin Feilberg, Taylor Connolly, Jillian St. George, Calen Ryan, Kaitlin Downs, Melanie Fetterly, Alex Yeo, Daniel Enright, Lindsay Bristow and Lynne Reykdal; laboratory crew: Darcy Childs, Michelle Ewacha and Calen Ryan. We also acknowledge the help of all the staff and volunteers at the Assiniboine Park Zoo. The authors thank two anonymous reviewers and Dr Rudy Boonstra, whose comments greatly improved this manuscript. Erika Mudrak and others at the Cornell Statistical Consulting Unit provided advice for the statistical analyses. Finally, we thank the 'Departed *Urocitellus* Dams' for the opportunity to bring this project to fruition.

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
