## [Reviewer comments · Royal Society Open Science]

Review History

RSOS-190904.R0 (Original submission)

Review form: Reviewer 1

Is the manuscript scientifically sound in its present form?

Yes

Are the interpretations and conclusions justified by the results?

Yes

Is the language acceptable?

Yes

Is it clear how to access all supporting data?

Yes

Do you have any ethical concerns with this paper?

No

Have you any concerns about statistical analyses in this paper?

No

Recommendation?

Accept with minor revision (please list in comments)

Comments to the Author(s)

Summary:

The authors analysed the effect of mother removal on post-weaning offspring in Richardson's ground squirrels, more specifically on their stress levels and survival. They further tested whether such removals affected neighbouring individuals (related or unrelated). They found no evidence that the removals affected either offspring, or the neighbours.

Overall:

I enjoyed the presented study very much and think it is of interest for a wide readership and would make a nice contribution. I think it is a very well written manuscript, that nicely reviews the literature in the introduction and makes clear a priori hypothesis and predictions. The fact the authors found no effect of removal does in no way take away from the study. I think negative results should be published more often in general.

My main issues concern the statistical analyses, or rather the presentation of the analyses. The text in the results section contains a lot of information in parentheses, which makes it hard to read. I think it would be nice if the authors would plot the estimates from their model (with standard error, or 95% Confidence intervals), which would make it much easier to see the strength and direction of the effects they found or the lack thereof. Alternatively the details about the F values and the non-significance could have been summarised in a table, as they did for one of the models (table 4).

Minor comments:

Line 150: I liked the very clearly stated hypotheses and predictions, however I think it is not necessary to present them both in text form, as well as a bullet list.

Lines 263-265: I was surprised, that the individual identity (removed individual) was not nested within the neighbourhood identity. Could the same animal be removed in different neighbourhoods, or why was this not a nested effect?

Line 269: Years are often treated as random factors in mixed model. Is there a particular reason why they were instead included as a fixed factor?

271-274: I would have liked a bit more information about how the coefficient of relatedness was calculated. Was it based on a pedigree did they use genetic marker to estimate the pairwise relatedness between individuals?

Lines 282 and onwards: Was there any type of model selection done, or was always the full model used in the end? And have the binomial GLMMs been tested for overdispersion?

Table 3: I'm not entirely sure how to interpret this table. There is no intercept, but the values look like taken directly from a typical model output. I assume that for example Proportion removed - 0.848 represents the slope for this factor and kin removed is the difference to the intercept, which would be kin not removed. I think it would be better to include the intercept, or alternatively plot (or give in a table) the back-transformed model estimates and their corresponding 95% CI for all the levels of a factor.

Fig 1: I think this information could instead be given in a table, which might even be easier to read and interpret.

Review form: Reviewer 2

Is the manuscript scientifically sound in its present form?

No

Are the interpretations and conclusions justified by the results?

No

Is the language acceptable?

Yes

Is it clear how to access all supporting data?

Yes

Do you have any ethical concerns with this paper?

No

Have you any concerns about statistical analyses in this paper?

No

Recommendation?

Major revision is needed (please make suggestions in comments)

Comments to the Author(s)

GENERAL COMMENTS

This is a fine study in which authors used a field removal approach on Richardson's ground squirrels to test two potentially related hypotheses. On the one hand, authors removed focal females that just breed successfully to test the possibility that the presence of mothers enhance reproductive success and survival of their weaned offspring. Thus, these effects were examined in the weaned offspring whose mother was removed and compared it with weaned offspring whose mother was not removed. Additionally, fecal metabolites of cortisol were also quantified to the offspring to get insights into the physiological mechanisms potentially underlying these putative effects. Secondly, authors also used this removal approach to examine whether the presence of kin neighbors had a positive influence on these same fitness and physiological responses. It turned out that none of these predicted effects were detected, implying that post-weaned females are resilient to drastic alterations of the maternal and social environment in the neighborhood. Overall, the field and analytical procedures used seemed appropriate to me. I particularly liked the upfront hypothesis-predictions "mode" used, an approach that was philosophically consistent with the specific model testing approach that followed.

I do have a few comments/suggestions to offer. In short, I am asking you to consider a more precise definition of sociality, especially when it comes to your model species. It seems to me that the current framework is used a bit loosely. This is problematic to fully understand your negative findings. Some aspects in your methods and terms used also require some clarifications or greater precision.

SPECIFIC COMMENTS

1. A main point I would ask authors to address is to better explain what is understood by sociality and then what they mean by a social species. From the initial paragraphs, it seems that sociality would characterize species that (i) form close (distinctive) social groups, where (ii) individuals attain fitness benefits despite some costs. Additionally, authors add the occurrence of cohesive behavior (iii) and delayed dispersal. I generally agree with these statements, however, we already know that net fitness benefits sometimes materialize in context-dependent ways where both ecological and social conditions play some roles. Regarding cohesive behavior, I am not sure what is meant here; are you implying situations in which group members carry their activities in a spatially cohesive way? Finally, evidence supporting that delayed dispersal is a condition for group-living to evolve remains equivocal.

2. The previous issue remains important for readers to understand how your model species fits or exhibits these attributes. Specifically, I would agree to that coloniality might be considered a form of sociality. However, the nature of social interactions among same sex individuals or between male-female breeding pairs in colonial species remain mostly neutral or even agonistic. Understanding this in your study species remains critical for readers to appreciate if we indeed should expect former maternal and current social (neighborhood) conditions to influence fitness and cort levels in the weaned offspring in the first place.

Thus, you should address in what sense Richardson's ground squirrel are indeed social. For instance, is there any evidence that females (excluding mother-dependent offspring) share their burrows, share their territories, or cooperate in to some context? You mentioned that females alarm call nepotistically, yet I always have wondered about how non kin also benefit similarly compared with kin. These animals, as other social and colonial rodents, move around to a good extent, so opportunities for non kin to "parasite" information from alarm callers is probably not that limited.

3. This study is aimed to understand how changes in the social environment experienced by weaned individuals may determine consequences. Specifically, authors focused their attention on fitness (reproductive success, survival) and physiology (glucocorticoids). I agree to that this addresses a relevant question. However, it seems to me that some work is needed to show a clearer picture about what we know, what is lacking, and then how this study adds to fill some of these gaps. Towards this end, I suggest emphasizing field over lab based evidence. Herein, you also would like to add some relatively recent field studies from group-living rodents, some of which also are colonial, documenting fitness effects as a consequence of changes in social conditions (e.g., Lardy et al. 2015 Ecology; Ebensperger et al. 2016 J Anim Ecol).

In this context I would like to make a suggestion on your use of the previous studies when making species contrasts to support your arguments. Through your Introduction and then through your Discussion you seem to mix scarcely (e.g., woodrats, meadow voles), moderately? (red deer), and highly social (black-tailed prairie dogs, tuco-tucos) species. So, I suggest a greater effort to contrast similarly social (or similarly asocial) species.

4. In my view, findings from this study remain interesting, but the evolutionary meaning of these remain a bit unclear. You explain your negative results based on high predation rate, a factor driving variation in social conditions. However, your findings also may reflect deeper rooted differences between highly social and scarcely social forms. In other words, these negative results might be what we should expect in a scenario in which social conditions play little if any role influencing fitness of a scarcely social rodent (compared with highly social counterpart). Under this scenario, your study may still contribute to shed lights into the evolved mechanisms underlying variation in sociality.

5. Regarding your methods, please state more explicitly if your protocol to quantify fecal cort metabolites was previously validated in your model species. Secondly, and I am sorry if I missed this completely, but I did not see how you calculated genetic relatedness among the females. Did you use DNA markers? If pedigrees were built based on trapping entire litters I wonder how multiple mating may have been an issue here.

6. Statistical analyses: in some of your models, "year" was entered as a fixed factor (bottom of page 11). This sounded a bit odd to me given that no specific interest was declared on each particular year of the study?

OTHER MINOR SUGGESTIONS

7. Title: I suggest a more process-oriented title. For instance, consider something like this: "Gone girl: offspring and neighbours are resilient to changes in their post-weaning maternal and other kin environment"??

8. I suggest using "annual reproductive success" instead of "reproduction" throughout the manuscript.

Decision letter (RSOS-190904.R0)

19-Jun-2019

Dear Dr Freeman,

The editors assigned to your paper ("Gone girl: Richardson's ground squirrel offspring and neighbours are resilient to female removal") have now received comments from reviewers. We would like you to revise your paper in accordance with the referee and Associate Editor suggestions which can be found below (not including confidential reports to the Editor). Please note this decision does not guarantee eventual acceptance.

Please submit a copy of your revised paper before 12-Jul-2019. Please note that the revision deadline will expire at 00.00am on this date. If we do not hear from you within this time then it will be assumed that the paper has been withdrawn. In exceptional circumstances, extensions may be possible if agreed with the Editorial Office in advance. We do not allow multiple rounds of revision so we urge you to make every effort to fully address all of the comments at this stage. If deemed necessary by the Editors, your manuscript will be sent back to one or more of the original reviewers for assessment. If the original reviewers are not available, we may invite new reviewers.

- Data accessibility

If you wish to submit your supporting data or code to Dryad (<http://datadryad.org/>), or modify your current submission to dryad, please use the following link:
<http://datadryad.org/submit?journalID=RSOS&manu=RSOS-190904>

- Competing interests

- Authors' contributions

- Acknowledgements

- Funding statement

on behalf of Dr Claudia Wascher (Associate Editor) and Kevin Padian (Subject Editor)
openscience@royalsociety.org

Associate Editor's comments (Dr Claudia Wascher):

The authors investigate how the loss of a mother affects juvenile Richardson's ground squirrels' stress hormones and survival. No impact on offspring or neighbouring conspecific glucocorticoid metabolites was described in the removal year, and no effect on overwinter survival in the following year. Both reviewers find the study interesting, however they also ask to elaborate on your definition of sociality, some clarification of your methods and presentation of results. Additional to the reviewers' comments, I have some further comments regarding the measurement of cortisol metabolites from faeces: Please clarify throughout the manuscript that you investigate glucocorticoid metabolites from faeces and not glucocorticoids. Please provide more details about sample collection, e.g. have all samples been collected at the same time of the day?, how many samples have been collected from how many individuals?, have fresh samples been collected from focal individuals or have samples been collected opportunistically in the territories?

Comments to Author:

Reviewers' Comments to Author:
Reviewer: 1

Comments to the Author(s)
Summary:

The authors analysed the effect of mother removal on post-weaning offspring in Richardson's ground squirrels, more specifically on their stress levels and survival. They further tested whether such removals affected neighbouring individuals (related or unrelated). They found no evidence that the removals affected either offspring, or the neighbours.

Overall:

I enjoyed the presented study very much and think it is of interest for a wide readership and would make a nice contribution. I think it is a very well written manuscript, that nicely reviews the literature in the introduction and makes clear a priori hypothesis and predictions. The fact the authors found no effect of removal does in no way take away from the study. I think negative results should be published more often in general.

My main issues concern the statistical analyses, or rather the presentation of the analyses. The text in the results section contains a lot of information in parentheses, which makes it hard to read. I think it would be nice if the authors would plot the estimates from their model (with standard error, or 95% Confidence intervals), which would make it much easier to see the

strength and direction of the effects they found or the lack thereof. Alternatively the details about the F values and the non-significance could have been summarised in a table, as they did for one of the models (table 4).

Minor comments:

Line 150: I liked the very clearly stated hypotheses and predictions, however I think it is not necessary to present them both in text form, as well as a bullet list.

Lines 263-265: I was surprised, that the individual identity (removed individual) was not nested within the neighbourhood identity. Could the same animal be removed in different neighbourhoods, or why was this not a nested effect?

Line 269: Years are often treated as random factors in mixed model. Is there a particular reason why they were instead included as a fixed factor?

271-274: I would have liked a bit more information about how the coefficient of relatedness was calculated. Was it based on a pedigree did they use genetic marker to estimate the pairwise relatedness between individuals?

Lines 282 and onwards: Was there any type of model selection done, or was always the full model used in the end? And have the binomial GLMMs been tested for overdispersion?

Table 3: I'm not entirely sure how to interpret this table. There is no intercept, but the values look like taken directly from a typical model output. I assume that for example Proportion removed - 0.848 represents the slope for this factor and kin removed is the difference to the intercept, which would be kin not removed. I think it would be better to include the intercept, or alternatively plot (or give in a table) the back-transformed model estimates and their corresponding 95% CI for all the levels of a factor.

Fig 1: I think this information could instead be given in a table, which might even be easier to read and interpret.

Reviewer: 2

Comments to the Author(s)
GENERAL COMMENTS

This is a fine study in which authors used a field removal approach on Richardson's ground squirrels to test two potentially related hypotheses. On the one hand, authors removed focal females that just breed successfully to test the possibility that the presence of mothers enhance reproductive success and survival of their weaned offspring. Thus, these effects were examined in the weaned offspring whose mother was removed and compared it with weaned offspring whose mother was not removed. Additionally, fecal metabolites of cortisol were also quantified to the offspring to get insights into the physiological mechanisms potentially underlying these putative effects. Secondly, authors also used this removal approach to examine whether the presence of kin neighbors had a positive influence on these same fitness and physiological responses. It turned out that none of these predicted effects were detected, implying that post-weaned females are resilient to drastic alterations of the maternal and social environment in the neighborhood. Overall, the field and analytical procedures used seemed appropriate to me. I particularly liked the upfront hypothesis-predictions "mode" used, an approach that was philosophically consistent with the specific model testing approach that followed.

I do have a few comments/suggestions to offer. In short, I am asking you to consider a more precise definition of sociality, especially when it comes to your model species. It seems to me that the current framework is used a bit loosely. This is problematic to fully understand your negative findings. Some aspects in your methods and terms used also require some clarifications or greater precision.

SPECIFIC COMMENTS

1. A main point I would ask authors to address is to better explain what is understood by sociality and then what they mean by a social species. From the initial paragraphs, it seems that sociality would characterize species that (i) form close (distinctive) social groups, where (ii) individuals attain fitness benefits despite some costs. Additionally, authors add the occurrence of cohesive behavior (iii) and delayed dispersal. I generally agree with these statements, however, we already know that net fitness benefits sometimes materialize in context-dependent ways where both ecological and social conditions play some roles. Regarding cohesive behavior, I am not sure what is meant here; are you implying situations in which group members carry their activities in a spatially cohesive way? Finally, evidence supporting that delayed dispersal is a condition for group-living to evolve remains equivocal.

2. The previous issue remains important for readers to understand how your model species fits or exhibits these attributes. Specifically, I would agree to that coloniality might be considered a form of sociality. However, the nature of social interactions among same sex individuals or between male-female breeding pairs in colonial species remain mostly neutral or even agonistic. Understanding this in your study species remains critical for readers to appreciate if we indeed should expect former maternal and current social (neighborhood) conditions to influence fitness and cort levels in the weaned offspring in the first place. Thus, you should address in what sense Richardson's ground squirrel are indeed social. For instance, is there any evidence that females (excluding mother-dependent offspring) share their burrows, share their territories, or cooperate in to some context? You mentioned that females alarm call nepotistically, yet I always have wondered about how non kin also benefit similarly compared with kin. These animals, as other social and colonial rodents, move around to a good extent, so opportunities for non kin to "parasite" information from alarm callers is probably not that limited.

3. This study is aimed to understand how changes in the social environment experienced by weaned individuals may determine consequences. Specifically, authors focused their attention on fitness (reproductive success, survival) and physiology (glucocorticoids). I agree to that this addresses a relevant question. However, it seems to me that some work is needed to show a clearer picture about what we know, what is lacking, and then how this study adds to fill some of these gaps. Towards this end, I suggest emphasizing field over lab based evidence. Herein, you also would like to add some relatively recent field studies from group-living rodents, some of which also are colonial, documenting fitness effects as a consequence of changes in social conditions (e.g., Lardy et al. 2015 Ecology; Ebensperger et al. 2016 J Anim Ecol). In this context I would like to make a suggestion on your use of the previous studies when making species contrasts to support your arguments. Through your Introduction and then through your Discussion you seem to mix scarcely (e.g., woodrats, meadow voles), moderately? (red deer), and highly social (black-tailed prairie dogs, tuco-tucos) species. So, I suggest a greater effort to contrast similarly social (or similarly asocial) species.

4. In my view, findings from this study remain interesting, but the evolutionary meaning of these remain a bit unclear. You explain your negative results based on high predation rate, a factor driving variation in social conditions. However, your findings also may reflect deeper rooted

differences between highly social and scarcely social forms. In other words, these negative results might be what we should expect in a scenario in which social conditions play little if any role influencing fitness of a scarcely social rodent (compared with highly social counterpart). Under this scenario, your study may still contribute to shed lights into the evolved mechanisms underlying variation in sociality.

5. Regarding your methods, please state more explicitly if your protocol to quantify fecal cort metabolites was previously validated in your model species. Secondly, and I am sorry if I missed this completely, but I did not see how you calculated genetic relatedness among the females. Did you use DNA markers? If pedigrees were built based on trapping entire litters I wonder how multiple mating may have been an issue here.

6. Statistical analyses: in some of your models, "year" was entered as a fixed factor (bottom of page 11). This sounded a bit odd to me given that no specific interest was declared on each particular year of the study?

OTHER MINOR SUGGESTIONS

7. Title: I suggest a more process-oriented title. For instance, consider something like this: "Gone girl: offspring and neighbours are resilient to changes in their post-weaning maternal and other kin environment"??

8. I suggest using "annual reproductive success" instead of "reproduction" throughout the manuscript.

Author's Response to Decision Letter for (RSOS-190904.R0)

See Appendix A.

Decision letter (RSOS-190904.R1)

23-Jul-2019

Dear Dr Freeman,

I am pleased to inform you that your manuscript entitled "Gone girl: Richardson's ground squirrel offspring and neighbours are resilient to female removal" is now accepted for publication in Royal Society Open Science.

Royal Society Open Science operates under a continuous publication model (<http://bit.ly/cpFAQ>). Your article will be published straight into the next open issue and this

will be the final version of the paper. As such, it can be cited immediately by other researchers. As the issue version of your paper will be the only version to be published I would advise you to check your proofs thoroughly as changes cannot be made once the paper is published.

on behalf of Dr Claudia Wascher (Associate Editor) and Kevin Padian (Subject Editor)
openscience@royalsociety.org

Appendix A

July 11, 2019

Dr. Claudia Wascher

Associate Editor, Royal Society Open Science

To the Editor and Reviewers:

We kindly thank both Reviewers and the Editor for their comments and suggestions regarding our manuscript. We have included additional information from highly-social species regarding social environment changes, and revised our presentation of our statistical results for ease of interpretation. Our specific responses, and the associated locations of each change are summarized below. We hope that you will now find our manuscript suitable for publication in Royal Society Open Science.

Kind regards,

Dr. Angela Freeman

Associate Editor's comments (Dr Claudia Wascher):

The authors investigate how the loss of a mother affects juvenile Richardson's ground squirrels' stress hormones and survival. No impact on offspring or neighbouring conspecific glucocorticoid metabolites was described in the removal year, and no effect on overwinter survival in the following year. Both reviewers find the study interesting, however they also ask to elaborate on your definition of sociality, some clarification of your methods and presentation of results. Additional to the reviewers' comments, I have some further comments regarding the measurement of cortisol metabolites from faeces: Please clarify throughout the manuscript that you investigate glucocorticoid metabolites from faeces and not glucocorticoids.

We have revised the manuscript throughout to indicate that our measurements were of fecal glucocorticoid metabolites, as suggested.

Please provide more details about sample collection, e.g. have all samples been collected at the same time of the day?,

Sampling time was restricted to times when squirrels are active and willing to enter traps, such that samples could be collected at any time during daylight trapping hours. We have revised our methods to indicate timing.

“All fecal samples were collected from 08:00 to 15:00 CDT, with variation in timing due to above-ground activity, and willingness of individuals squirrels to enter live-traps. We collected fresh feces within 30 minutes of defecation using clean, disposable wooden sticks. Only feces uncontaminated with urine were collected.”

Lines 247-251

how many samples have been collected from how many individuals?,

We collected 414 samples from 143 individual squirrels, and directly mention this in our results.

“We collected a total of 414 fecal samples from 143 individual squirrels.”

Lines 353-354

have fresh samples been collected from focal individuals or have samples been collected opportunistically in the territories?

We collected fecal samples by identifying neighbor kin and non-kin, and offspring of female subjects and attempting to sample from these individuals with targeted live-trapping. For controls we collected as many samples from as many individuals as possible since we could not know during collection who would survive until the end of season. We have clarified this in the methods section.

“We collected *ad hoc* fecal samples from neighbours and offspring in the population throughout the summer, sampling from neighbours adjacent to where females were being removed. Once we confirmed which females had survived late into the season, we randomly selected a female in the same neighbourhood as the associated control.”

Lines 256-264.

Comments to Author:

Reviewers' Comments to Author:

Reviewer: 1

Comments to the Author(s)

Summary:

The authors analysed the effect of mother removal on post-weaning offspring in Richardson's ground squirrels, more specifically on their stress levels and survival. They further tested whether such removals affected neighbouring individuals (related or unrelated). They found no

evidence that the removals affected either offspring, or the neighbours.

Overall:

I enjoyed the presented study very much and think it is of interest for a wide readership and would make a nice contribution. I think it is a very well written manuscript, that nicely reviews the literature in the introduction and makes clear a priori hypothesis and predictions. The fact the authors found no effect of removal does in no way take away from the study. I think negative results should be published more often in general.

We thank the reviewer for their support and comments.

My main issues concern the statistical analyses, or rather the presentation of the analyses. The text in the results section contains a lot of information in parentheses, which makes it hard to read. I think it would be nice if the authors would plot the estimates from their model (with standard error, or 95% Confidence intervals), which would make it much easier to see the strength and direction of the effects they found or the lack thereof. Alternatively the details about the F values and the non-significance could have been summarised in a table, as they did for one of the models (table 4).

We thank the reviewer for the suggestions to improve the readability of the results. As suggested we have presented the model summary statistics in tables 3, 5, and 6 (and in the supplement). In the text, we have included the effect sizes, actual means and SDs for the data when the model estimates were included in tables. For data in supplementary tables (i.e. analyses of datasets from the 3-to-5 day restricted window post-removals), we included the same information in the main manuscript, though confidence intervals and other model summary statistics are in the supplement.

Minor comments:

Line 150: I liked the very clearly stated hypotheses and predictions, however I think it is not necessary to present them both in text form, as well as a bullet list.

We have included both because the overarching hypothesis has 4 parts (listed in bullets) and our overarching prediction was similarly associated with multiple sub-predictions based on the associated hypotheses.

Lines 263-265: I was surprised, that the individual identity (removed individual) was not nested within the neighbourhood identity. Could the same animal be removed in different neighbourhoods, or why was this not a nested effect?

We thank the reviewer for the suggestion – indeed they should be listed as nested effects since each female was only a member of one neighbourhood. We have revised our analyses accordingly, though our interpretation and findings have not changed.

Line 269: Years are often treated as random factors in mixed model. Is there a particular reason why they were instead included as a fixed factor?

Since we only had two years of data, we treated year as a fixed factor as our understanding is that for random factors to be effective in the model there need to be more than five levels

(Crawley 2002; The R book, page 473).

271-274: I would have liked a bit more information about how the coefficient of relatedness was calculated. Was it based on a pedigree did they use genetic marker to estimate the pairwise relatedness between individuals?

We thank the reviewer for the suggestion – we assumed littermates to be full siblings, with a coefficient of relatedness of 0.5 based on maternal pedigree. There were two reasons for this approach. 1. At the time of designing this study, we only had known matrilineal relatedness in the colony and not genetic relatedness. 2. While multiple paternity does exist in our species, these ground squirrels discriminate all littermates from non-littermate neighbours, and non-littermate strangers..

“Individuals from a single litter were assigned a coefficient of relatedness of 0.5, given we had only maternal relatedness data from our pedigree records, and because juveniles behaviorally discriminate littermates from non-littermates, even though some individuals were undoubtedly maternal half-siblings due to multiple paternity (44,45).”

Line 215-219

Lines 282 and onwards: Was there any type of model selection done, or was always the full model used in the end? And have the binomial GLMMs been tested for overdispersion?

We used the full model in all cases, and have tested our GLMMs for overdispersion (none violated assumptions).

“Binomial and Poisson models were tested for overdispersion using the Chi-square method for the observed and predicted variance (53).”

Lines 342-343

Table 3: I'm not entirely sure how to interpret this table. There is no intercept, but the values look like taken directly from a typical model output. I assume that for example Proportion removed -0.848 represents the slope for this factor and kin removed is the difference to the intercept, which would be kin not removed. I think it would be better to include the intercept, or alternatively plot (or give in a table) the back-transformed model estimates and their corresponding 95% CI for all the levels of a factor.

We regret that we made an error excluding the intercept in this table. In addition to revising this table (to include the reference group), we have also included the Wald 95% confidence intervals. For other data, to reduce the amount of information in parentheses (as noted above), we opted to present these data in tables as well. We also included the same information regarding parameter estimates and confidence intervals in the supplementary data in tables.

Fig 1: I think this information could instead be given in a table, which might even be easier to read and interpret.

We thank the reviewer for the suggestion, and have included these data as a table. (Table 2).

Reviewer: 2

Comments to the Author(s)

GENERAL COMMENTS

This is a fine study in which authors used a field removal approach on Richardson's ground squirrels to test two potentially related hypotheses. On the one hand, authors removed focal females that just breed successfully to test the possibility that the presence of mothers enhance reproductive success and survival of their weaned offspring. Thus, these effects were examined in the weaned offspring whose mother was removed and compared it with weaned offspring whose mother was not removed. Additionally, fecal metabolites of cortisol were also quantified to the offspring to get insights into the physiological mechanisms potentially underlying these putative effects. Secondly, authors also used this removal approach to examine whether the presence of kin neighbors had a positive influence on these same fitness and physiological responses. It turned out that none of these predicted effects were detected, implying that post-weaned females are resilient to drastic alterations of the maternal and social environment in the neighborhood. Overall, the field and analytical procedures used seemed appropriate to me. I particularly liked the upfront hypothesis-predictions "mode" used, an approach that was philosophically consistent with the specific model testing approach that followed.

I do have a few comments/suggestions to offer. In short, I am asking you to consider a more precise definition of sociality, especially when it comes to your model species. It seems to me that the current framework is used a bit loosely. This is problematic to fully understand your negative findings. Some aspects in your methods and terms used also require some clarifications or greater precision.

SPECIFIC COMMENTS

1. A main point I would ask authors to address is to better explain what is understood by sociality and then what they mean by a social species. From the initial paragraphs, it seems that sociality would characterize species that (i) form close (distinctive) social groups, where (ii) individuals attain fitness benefits despite some costs. Additionally, authors add the occurrence of cohesive behavior (iii) and delayed dispersal. I generally agree with these statements, however, we already know that net fitness benefits sometimes materialize in context-dependent ways where both ecological and social conditions play some roles. Regarding cohesive behavior, I am not sure what is meant here; are you implying situations in which group members carry their activities in a spatially cohesive way? Finally, evidence supporting that delayed dispersal is a condition for group-living to evolve remains equivocal.

We agree with the reviewer that context and life-history do impact the expression of 'sociality' in a broad sense. By 'cohesive' behavior, we are referring to the social greeting, grooming, and

affiliation and reduced agonistic behavior. We have clarified this on Line 68. While this can be spatially cohesive, it is not necessarily so.

We do not think that delayed dispersal is necessary for sociality to evolve, but it is associated with an increased likelihood, particularly for the ground-dwelling squirrels. In Richardson's, female kin (typically yearlings with their mother, or yearlings together) will cohabitate in the early spring. (Lines 134-138).

2. The previous issue remains important for readers to understand how your model species fits or exhibits these attributes. Specifically, I would agree to that coloniality might be considered a form of sociality. However, the nature of social interactions among same sex individuals or between male-female breeding pairs in colonial species remain mostly neutral or even agonistic. Understanding this in your study species remains critical for readers to appreciate if we indeed should expect former maternal and current social (neighborhood) conditions to influence fitness and cort levels in the weaned offspring in the first place.

Thus, you should address in what sense Richardson's ground squirrel are indeed social. For instance, is there any evidence that females (excluding mother-dependent offspring) share their burrows, share their territories, or cooperate in to some context? You mentioned that females alarm call nepotistically, yet I always have wondered about how non kin also benefit similarly compared with kin. These animals, as other social and colonial rodents, move around to a good extent, so opportunities for non kin to "parasite" information from alarm callers is probably not that limited.

To address the level of sociality, we have included additional information about how females share space. Indeed, much of the sociality in Richardson's is driven by kinship established by close association. We have included additional information on Lines 134-138 regarding the level of sociality.

"After emerging from hibernation in the spring but before giving birth to young, adult female kin will often share sleeping burrows (38), though there is no evidence that they nest with or care for (i.e. alloparent or allonurse) the pre-weaned offspring of kin. Throughout the season, kin will overlap in area use, and adult females are more tolerant of close relatives (39,40)."

It is likely, however, that unrelated females surrounded by a kin-cluster could benefit by eavesdropping on females engaged in kin-directed nepotistic behavior. To our knowledge, this has not been directly tested. To address this, we added on lines 184-185:

"We made similar predictions for neighbours: if individuals are obtaining benefits from their neighbours (e.g. through social stability, alarm calling, kinship, reciprocity or byproduct mutualism), we should see increased fecal glucocorticoid metabolites (H3) and reduced survival and reproduction of neighbours when others are removed (H4)."

3. This study is aimed to understand how changes in the social environment experienced by weaned individuals may determine consequences. Specifically, authors focused their attention on fitness (reproductive success, survival) and physiology (glucocorticoids). I agree to that this addresses a relevant question. However, it seems to me that some work is needed to show a clearer picture about what we know, what is lacking, and then how this study adds to fill some of these gaps. Towards this end, I suggest emphasizing field over lab based evidence.

Herein, you also would like to add some relatively recent field studies from group-living rodents, some of which also are colonial, documenting fitness effects as a consequence of changes in social conditions (e.g., Lardy et al. 2015 Ecology; Ebensperger et al. 2016 J Anim Ecol). In this context I would like to make a suggestion on your use of the previous studies when making species contrasts to support your arguments.

Through your Introduction and then through your Discussion you seem to mix scarcely (e.g., woodrats, meadow voles), moderately? (red deer), and highly social (black-tailed prairie dogs, tuco-tucos) species. So, I suggest a greater effort to contrast similarly social (or similarly asocial) species.

We thank the reviewer for the suggestion. We included these studies in a range of species, specifically because they examine the effects of removal of parents in the post-weaning period and how it influences non-reproductive offspring in later life. We noted in the introduction that the effects of social change are largely species and life-history specific (Lines 99-100).

“In addition to the impacts of changing parental care, changing an animal’s broader social environment can influence survival, reproductive fitness, and physiology. The direction of these effects are largely species and life-history specific and are influenced by myriad external and internal factors (e.g. population density, breeding state).”

We agree that we should include more information on adult individuals and their altered social environment. We have included additional information from these other suggested field studies in our introduction.

“In social species such as the Alpine marmot (*Marmota marmota*), and the degu (*Octodon degus*), changes in the social environment can impact reproductive success by altering intrasexual competition, and opportunities for cooperative offspring care, respectively (29,30).”

Lines 114-117

4. In my view, findings from this study remain interesting, but the evolutionary meaning of these remain a bit unclear. You explain your negative results based on high predation rate, a factor driving variation in social conditions. However, your findings also may reflect deeper rooted differences between highly social and scarcely social forms. In other words, these negative results might be what we should expect in a scenario in which social conditions play little if any role influencing fitness of a scarcely social rodent (compared with highly social counterpart). Under this scenario, your study may still contribute to shed lights into the evolved mechanisms underlying variation in sociality.

We agree that the level of sociality (highly social vs asocial or rarely social) could impact whether losing a member of the group influences individuals.

Certainly, how one defines ‘highly social’ will also impact the interpretation. In our discussion, we have now included the following in our discussion:

“While Richardson’s ground squirrels are social, live in large colonies, and benefit from shared defense and vigilance towards predators, they do not alloparent the offspring of kin, and males do not engage in

paternal care. Whether similar species which alloparent or are biparental have similar resilience to social change remains an open question; though evidence from two highly social species suggests that changes in the social environment might detrimentally impact the number of weaned offspring (30), and their survival (29).”

Lines 521-527

5. Regarding your methods, please state more explicitly if your protocol to quantify fecal cort metabolites was previously validated in your model species.

We thank the reviewer for the suggestion, and we have added this to the methods section.

Secondly, and I am sorry if I missed this completely, but I did not see how you calculated genetic relatedness among the females. Did you use DNA markers? If pedigrees were built based on trapping entire litters I wonder how multiple mating may have been an issue here.

We set littermates as ‘full siblings’ in terms of relationships due to the behavioral discrimination in this species between littermates and non-littermates, and the lack of historical genetic information. However, it is likely that some of the siblings were in fact, maternal half-siblings. (Lines 215-219)

“Individuals from a single litter were assigned a coefficient of relatedness of 0.5, given we had only maternal relatedness data from our pedigree records, and because juveniles behaviorally discriminate littermates from non-littermates, even though some individuals were undoubtedly maternal half-siblings due to multiple paternity (44,45).”

6. Statistical analyses: in some of your models, “year” was entered as a fixed factor (bottom of page 11). This sounded a bit odd to me given that no specific interest was declared on each particular year of the study?

While we had no a priori predictions concerning one year over another, we wanted to address the potential for seasonal variation in overall number of predators or food availability in the effects of social environment changes. We have included a statement in the methods to address this. Additionally, inclusion as a random effect is only recommended when there are > 5 levels within the factor (Crawley, the R book, page 473), which, we did not have for this variable (Lines 311-313)

“We included year as a fixed effect in our analyses to help account for differences in predator abundance, food availability, and weather between years.”

OTHER MINOR SUGGESTIONS

7. Title: I suggest a more process-oriented title. For instance, consider something like this: “Gone girl: offspring and neighbours are resilient to changes in their post-weaning maternal and other kin environment”??

We have spent quite a bit of time thinking about the title. Because we tested both offspring and neighbours, a title which tries to include statements about both groups can get quite long. As such we kept it simple, and opted for inclusion in the keywords for search optimization.

8. I suggest using “annual reproductive success” instead of “reproduction” throughout the manuscript.

Where this makes sense to do so, we have changed the wording. In some cases, we are referring to both the reproductive success and reproductive timing, and have left the phrasing as-is. (Line 166)